# BAYES CLASSIFIER CANNOT BE LEARNED FROM NOISY RESPONSES WITH UNKNOWN NOISE RATES

**Soham Bakshi***
Department of Statistics
University of Michigan
baksho@umich.edu

**Subha Maity***
Department of Statistics
University of Michigan
smaity@umich.edu

## ABSTRACT

Training a classifier with noisy labels typically requires the learner to specify the distribution of label noise, which is often unknown in practice. Although there have been some recent attempts to relax that requirement, we show that the Bayes decision rule is unidentified in most classification problems with noisy labels. This suggests it is generally not possible to bypass/relax the requirement. In the special cases in which the Bayes decision rule is identified, we develop a simple algorithm to learn the Bayes decision rule, that does not require knowledge of the noise distribution.

## 1 INTRODUCTION

In this paper, we consider classification with noisy labels. Let $\mathcal{X}$ and $\mathcal{Y}$ be the feature/input and label/output spaces, respectively. The clean/noiseless samples $(X_i, Y_i)$ are drawn independently from $P_{X,Y} \in \Delta(\mathcal{X} \times \mathcal{Y})$ ($\Delta(\mathcal{A})$ is the space of probability measures on $\mathcal{A}$), but the learner only observes the $(X_i, Y_i')$'s, where $Y_i'$ is corrupted version of $Y_i$ from a conditional distribution: $Y' \mid Y \sim P_{Y'|Y}$. The learner seeks to estimate the Bayes classifier of $Y$ (the clean/noiseless label)

$$f_P^\star(x) \triangleq \arg\max_{y \in \mathcal{Y}} P(Y = y \mid X = x)$$

from the noisy training data $\{(X_i, Y_i')\}_{i=1}^n$. The classification with noisy labels problem arises in many areas of science and engineering, including medical image analysis (Karimi et al., 2020) and crowdsourcing (Jiang et al., 2021).

When the label noise rates/distribution $P_{Y'|Y}$ is known or learnable from external data, there are several methods to recover $f_P^\star$ (Bylander, 1994; Cesa-Bianchi et al., 1999; Natarajan et al., 2013). Unfortunately, $P_{Y'|Y}$ is often unknown to the learner in practice, which limits their applicability. Recently, Liu & Guo (2020) propose a method based on *peer prediction*, which provably recovers $f_P^\star$ when there are only two classes and they are balanced (but the label noise distribution is unknown). The framework of peer prediction is motivated from Dasgupta & Ghosh (2013); Shnayder et al. (2016), and has been further developed for ranking problems with noisy labels (Wu et al., 2022). A review on learning from noisy labels and peer prediction can be found in Appendix A.

In this paper, we consider the statistical aspects of classification with noisy labels. Our main contributions are the following.

- We show that the balanced binary classification problem is the only instance in which $f_P^\star$ can be learned without knowledge of the label noise distribution, while in more general problems (with imbalanced or more than two classes), that knowledge is essential.

- We develop a new method based on weighted empirical risk minimization (ERM) that provably learns $f_P^\star$ in the balanced binary classification problem with noisy labels.

## 2 IDENTIFIABILITY OF THE BAYES CLASSIFIER

In our setup a typical data-point $(X, Y, Y')$ (a triplet of feature, clean label and noisy label) comes from a true distribution $P \equiv P_{X,Y,Y'}$, whose full joint distribution is unknown. Since the learner only observes iid $(X_i, Y_i')$ pairs we assume that the $P_{X,Y'}$ marginal is known. Furthermore, we assume that the noise rates/distributions are *instance independent*, *i.e.*, for any $x \in \mathcal{X}$ and $y, y' \in \mathcal{Y}$

$$P(Y' = y' \mid Y = y, X = x) = P(Y' = y' \mid Y = y) \triangleq \epsilon_P(y', y). \tag{1}$$

---

*Equal contributions.

In our investigation we fix the marginal of $Y$, *i.e.*, for some $p \in \Delta(\mathcal{Y})$ we assume that $P_Y = p$. Thus, we define the class $\mathcal{Q}(K, p)$ of all probabilities $Q \equiv Q_{X,Y,Y'} \in \Delta(\mathcal{X} \times \mathcal{Y} \times \mathcal{Y})$ that satisfy (1) $Q_{X,Y'} = P_{X,Y'}$, (2) has instance independent noise (*i.e.*, $Q$ satisfies equation 1), (3) $Q_Y = p$, and (4) the determinant of $E_Q = [[\epsilon_Q(y', y)]]_{y',y \in \mathcal{Y}}$ is positive. The final condition is a regularity condition on the noise rates and is satisfied if $Q(Y' \neq Y \mid Y = y)$ are not too large. In fact, for binary classification it boils down to $\epsilon_Q(0,1) + \epsilon_Q(1,0) < 1$, which is a rather weak assumption and standard in the literature (Natarajan et al., 2013; Liu & Guo, 2020).

In the following theorem, whose proof can be found in Appendix C, we investigate whether the Bayes classifier $f^\star_Q(x) \triangleq \arg\min_{y \in \mathcal{Y}} Q(Y = y \mid X = x)$ is same for all the $Q \in \mathcal{Q}(K, p)$.

**Theorem 2.1** (Identifiability of the Bayes classifier). *The Bayes classifier $f^\star_Q$ is unique for all $Q \in \mathcal{Q}(K, p)$, i.e., $\{f^\star_Q : Q \in \mathcal{Q}(K, p)\}$ is a singleton set if and only if $K = 2$ and $p = (1/2, 1/2)$.*

## 2.1 THE IDENTIFIABLE CASE

Balanced binary classification *i.e.*, $p = (1/2, 1/2)$ is the special case when *the Bayes classifier is unique, regardless of the noise rates*. This is the optimistic case where the noise rates are not required for learning $f^\star_P$. In this case *we provide an alternative* to the peer loss framework (Liu & Guo, 2020) for learning $f^\star_P$ that relies on the popular weighted ERM method:

$$\min_{\eta \in \mathcal{F}} \frac{1}{n} \sum_{i=1}^{n} p_n(1 - Y'_i)\ell(\eta(X_i), Y'_i), \tag{2}$$

where $\mathcal{F}$ is a set of probabilistic classification models such that for some $\eta^\star \in \mathcal{F}$ it holds $f^\star_P(x) = \mathbb{1}\{\eta^\star(x) \geq 1/2\}$, $\ell$ is an appropriate loss function, and $p_n(y) \triangleq 1/n \sum_{i=1}^{n} \mathbb{1}\{Y'_i = y\}$. The following lemma, whose proof can be found in Appendix E, establishes that the weighted ERM in equation 2 using the noisy distribution $(P_{X,Y'})$ estimates $f^\star_P$, regardless of the noise rates.

**Lemma 2.2** (Weighted ERM). *Let $Y, Y' \in \{0, 1\}$, $P(Y = 1) = 1/2$, $\mathcal{F}$ be the set of all binary classifiers on $\mathcal{X}$, $\ell(f(x), y) = \mathbb{1}\{f(x) \neq y\}$ for $f \in \mathcal{F}$ and $\epsilon_P(0,1) + \epsilon_P(1,0) < 1$. If $p'(1) = P(Y' = 1)$ and $p'(0) = P(Y' = 0)$ then regardless the values of $\epsilon_P(0,1)$ and $\epsilon_P(1,0)$ the Bayes classifier is recovered, i.e.*

$$f^\star_P(x) = \arg\min_{f \in \mathcal{F}} \mathbb{E}_P[p'(1 - Y')\ell(f(X), Y')]. \tag{3}$$

Thus, the weighted ERM is an alternative framework to the peer loss (Liu & Guo, 2020) for learning $f^\star_P$ from noisy labels. In Appendix D we compare these two frameworks, where we also highlight a drawback of the peer loss, that it may not be bounded below and may diverge to $-\infty$ while minimization. Our method, on the other hand, does not suffer from it.

Though the noise rates are not needed for the aforementioned case, it is impossible to verify whether $P(Y = 1) = 1/2$ if nothing is known about $P_Y$. So, the weighted ERM may not be practical without precise information about $P_Y$, which is also required by the peer loss framework.

## 2.2 THE NON-IDENTIFIABLE CASES

For imbalanced binary classification or with more than two classes the Bayes classifier is not identifiable when the noise rates ($P_{Y'|Y}$) are unknown. In fact, for establishing the proof of Theorem 2.1 we construct two different $P_{Y'|Y}$'s that are compatible with the marginals $P_{X,Y'}$ and $P_Y$ but have different Bayes decision boundaries. This is the problematic case, where it is statistically impossible to learn the Bayes classifier owing to lack of identifiability, and *an additional knowledge on $P_{Y'|Y}$ is essential for developing meaningful procedures.*

## 3 DISCUSSION

In this study, we present a thorough examination of the identifiability of the Bayes classifier in classification scenarios with noisy labels where the noise rates are unknown. The necessity of knowing the noise rates (in general) is clear from our results: in almost all cases, it is impossible to learn the Bayes classifier for the true labels without this piece of information. We hope that our findings can help practitioners develop a better understanding about the limitations and requirements for learning classification models from noisy labels.

URM STATEMENT

The authors acknowledge that both the authors of this work meet the URM criteria of ICLR 2023 Tiny Papers Track.

ACKNOWLEDGMENTS

The authors would like to thank Prof. Yuekai Sun and Prof. Moulinath Banerjee for their insightful comments and discussions related to this work. Subha Maity was supported by the National Science Foundation (NSF) under grants no. 2027737 and 2113373 while working on this paper.

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

## A    RELATED WORK

**Learning with noisy response** has been a topic of great importance many areas of science and engineering, including medical image analysis (Karimi et al., 2020) and crowdsourcing (Jiang et al., 2021). It has produced a wide range of researches related to importance re-weighting algorithm (Liu & Tao, 2015), robust cross-entropy loss for neural networks (Zhang & Sabuncu, 2018), loss correction (Patrini et al., 2017), learning noise rates (Liu & Tao, 2015; Patrini et al., 2017; Xiao et al., 2015). A more comprehensive reviews about the literature can be found in Song et al. (2022); Liang et al. (2022).

Our work considers the classification problems with noisy labels when the noise rates are **instance independent**, as specified in equation 1. When the noise rates are known, various methods for learning a classifier have been proposed by Bylander (1994); Cesa-Bianchi et al. (1999); Natarajan et al. (2013). Liu & Guo (2020) recently introduced the peer loss approach for learning a classifier for binary classification problem without prior knowledge of the noise rates. However, it remains unclear whether this approach can be extended beyond binary classification. Our work addresses this gap and complements the findings by Liu & Guo (2020) in two ways: (1) our results explain why a classification task can be performed for balanced binary classification problems without requiring knowledge of the noise rates, and (2) we demonstrate that this is the only scenario in which the Bayes classifier is uniquely identified and a statistically consistent Bartlett et al. (2006) classification is possible.

**Peer loss** has been proposed by Liu & Guo (2020) for learning the classifier without knowing the noise rates, which is most related to this work. The peer loss has been motivated from the ideas of peer prediction (Prelec, 2004; Miller et al., 2005; Witkowski & Parkes, 2012; Dasgupta & Ghosh, 2013; Witkowski et al., 2013; Radanovic et al., 2016; Shnayder et al., 2016), and has been further developed for ranking problems (Wu et al., 2022).

## B    SYNTHETIC EXPERIMENT

We empirically investigate the classification approach in equation 2 on a synthetic dataset for binary classification, whose description follows: (1) $Y \sim \text{Bernoulli}(p)$, (2) $X \mid (Y = y) \sim \text{N}_2(^1\!/_2(2y - 1)\mathbf{1}_2, \mathbf{I}_2)$, and (3) $P(Y' = 0 \mid Y = 1) = \epsilon_1$, $P(Y' = 1 \mid Y = 0)\epsilon_0$. We consider two situations: (1) the balanced case ($p = {}^1\!/_2$), when the Bayes classifier is identified, and (2) an imbalanced

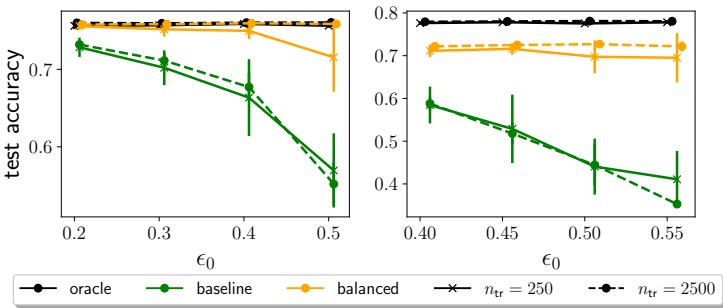

Figure 1: The consistency (resp. inconsistency) of class balancing approach in equation 2 for balanced (resp. imbalanced) binary classification, as observed in left (resp. right) plot.

case with $p = 0.35$, when the Bayes classifier is not identified. For both the cases we set $\epsilon_1 = \{0.4-(1-\epsilon_0)(1-p)\}/p$. Note that, for such a choice

$$P(Y' = 0) = P(Y' = 0 \mid Y = 1)p + P(Y' = 0 \mid Y = 0)(1 - p) = \epsilon_1 p + (1 - \epsilon_0)(1 - p) = 0.4 \,.$$

We compare the classification approach in equation 2 with two baselines: (1) the *oracle*, trained with the true $Y$, and (2) the *baseline*, trained the noisy $Y'$ without any adaptation. We use logistic regression model for all the classifiers. We do not consider peer-loss Liu & Guo (2020) as a baseline for its failure case described in Appendix D.

In the balanced case (left plot in Figure 1) we see that the class balancing approach in equation 2 has identical performance to the oracle/ideal case for large sample sizes ($n_{\text{tr}} = 2500$), which is an evidence for it's statistical consistency Lugosi & Vayatis (2004). On contrary, for imbalanced case (right plot in the same figure) there is a gap between the class balancing approach and the oracle, even for large sample sizes, *i.e.*, the class balancing approach is statistically inconsistent.

## C  Proof of Theorem 2.1

For readers convenience we restate the Theorem 2.1.

**Theorem C.1** (Identifiability of the Bayes classifier). *The Bayes classifier $f_Q^\star$ is unique for all $Q \in \mathcal{Q}(K, p)$, i.e., $\{f_Q^\star : Q \in \mathcal{Q}(K, p)\}$ is a singleton set if and only if $K = 2$ and $p = (1/2, 1/2)$.*

For $Q \in \mathcal{Q}(K, p)$ we recall the definition of the Bayes classifier that

$$f_Q^\star(x) = \arg\max_{k \in \mathcal{Y}} Q(Y = k \mid X = x) = \arg\max_{k \in \mathcal{Y}} Q(Y = k \mid X = x) p_X(x)$$

where $p_X$ is the density function of $X$. Note that the marginal of $X$ are same for $P$ and $Q$. Henceforth, we define the class probabilities of $Y'$ as $Q_{Y'} = P_{Y'} = p'$,

$$\alpha_k(x) = Q_{X,Y}(x, k) \triangleq Q(Y = k \mid X = x) p_X(x),$$
$$a_k(x) = Q_{X,Y'}(x, k) \triangleq Q(Y' = k \mid X = x) p_X(x) \,.$$

We further recall that $\epsilon(k', k) = Q_{Y'|Y}(Y' = k' \mid Y = k)$.

### C.1  The binary classification case

Here we let $\mathcal{Y} = \{1, 2\}$ and establish that the Bayes decision boundary is unique if and only if $p = (1/2, 1/2)$. We begin with a lemma that we require for the proof.

**Lemma C.2.** *The following holds*

$$\begin{bmatrix} \alpha_1(x) \\ \alpha_2(x) \end{bmatrix} = \frac{1}{1 - \epsilon(1, 2) - \epsilon(2, 1)} \begin{bmatrix} 1 - \epsilon(1, 2) & -\epsilon(1, 2) \\ -\epsilon(2, 1) & 1 - \epsilon(2, 1) \end{bmatrix} \begin{bmatrix} a_1(x) \\ a_2(x) \end{bmatrix} \,. \tag{4}$$

*Proof of Lemma C.2.* Let's start with $a_1(x)$, by definition $a_1(x) = Q_{(X,Y')}(x,1)$.

$$
\begin{aligned}
a_1(x) = Q_{(X,Y')}(x,1) &= Q_{(X,Y,Y')}(x,1,1) + Q_{(X,Y,Y')}(x,2,1) \\
&= Q_X(x)Q_{Y|X}(1|x)Q_{Y'|(X,Y)}(1|(x,1)) + Q_X(x)Q_{Y|X}(2|x)Q_{Y'|(X,Y)}(1|(x,2)) \\
&= Q_X(x)Q_{Y|X}(1|x)Q_{Y'|Y}(1|1) + Q_X(x)Q_{Y|X}(2|x)Q_{Y'|Y}(1|2) \\
&= Q_{(X,Y)}(x,1)Q_{Y'|Y}(1|1) + Q_{(X,Y)}(x,2)Q_{Y'|Y}(1|2) \\
&= (1 - \epsilon(2,1))\alpha_1(x) + \epsilon(1,2)\alpha_2(x),
\end{aligned}
$$

where in the third equality we use $X \perp Y' \mid Y$, that the noise rates are instance independent. Similarly for $a_2(x) = Q_{(X,Y')}(x,2)$ we have

$$
a_2(x) = \epsilon(2,1)\alpha_1(x) + (1 - \epsilon(1,2))\alpha_2(x).
$$

In matrix notation, we have $\begin{bmatrix} a_1(x) \\ a_2(x) \end{bmatrix} = \begin{bmatrix} 1 - \epsilon(2,1) & \epsilon(1,2) \\ \epsilon(2,1) & 1 - \epsilon(1,2) \end{bmatrix} \begin{bmatrix} \alpha_1(x) \\ \alpha_2(x) \end{bmatrix}$, which we invert to conclude the proof.

$$
\begin{bmatrix} \alpha_1(x) \\ \alpha_2(x) \end{bmatrix} = \frac{1}{1 - \epsilon(1,2) - \epsilon(2,1)} \begin{bmatrix} 1 - \epsilon(1,2) & -\epsilon(1,2) \\ -\epsilon(2,1) & 1 - \epsilon(2,1) \end{bmatrix} \begin{bmatrix} a_1(x) \\ a_2(x) \end{bmatrix}.
$$

$\square$

Now, $Q_Y = p$ whenever $Q \in \mathcal{Q}(2,p)$. Denoting $p_1 = Q(Y=1)$ and $p_1' = Q(Y'=1)$ we notice that

$$
\begin{aligned}
p_1' &= Q(Y'=1|Y=1)Q(Y=1) + Q(Y'=1|Y=2)Q(Y=2) \\
&= p_1(1 - \epsilon(2,1) + (1 - p_1)\epsilon(1,2),
\end{aligned}
$$

$$
\text{or,} \quad \epsilon(2,1) = \frac{p_1 - p_1' + \epsilon(1,2) - \epsilon(1,2)p_1}{p_1}.
$$

We now use the above equation to replace $\epsilon(2,1)$ in Lemma C.2. Notice that, according to Lemma C.2

$$
\alpha_1(x) - \alpha_2(x) = \frac{(1 - \epsilon(1,2) + \epsilon(2,1))a_1(x) - (1 - \epsilon(2,1) + \epsilon(2,1))a_2(x)}{1 - \epsilon(1,2) - \epsilon(2,1)}.
$$

We now plug in the expression of $\epsilon(2,1)$ in terms of $\epsilon(1,2)$, $p'$ and $p$, and obtain

$$
1 - \epsilon(1,2) + \epsilon(2,1) = \frac{p - p\epsilon(1,2) + p - p' + \epsilon(1,2) - \epsilon(1,2)p}{p} = 2 - \frac{p' + \epsilon(1,2)(2p - 1)}{p}
$$

$$
1 - \epsilon(2,1) + \epsilon(2,1) = \frac{p + p\epsilon(1,2) - p + p' - \epsilon(1,2) + \epsilon(1,2)p}{p} = \frac{p' + \epsilon(1,2)(2p - 1)}{p}.
$$

Thus,

$$
\begin{aligned}
&(1 - \epsilon(1,2) - \epsilon(2,1))(\alpha_1(x) - \alpha_2(x)) \\
&= (1 - \epsilon(1,2) + \epsilon(2,1))a_1(x) - (1 - \epsilon(2,1) + \epsilon(2,1))a_2(x) \\
&= 2a_1(x) - \frac{p' + \epsilon(1,2)(2p - 1)}{p}(a_1(x) + a_2(x))
\end{aligned}
\tag{5}
$$

Since $\epsilon(1,2) + \epsilon(2,1) < 1$ we have

$$
\begin{aligned}
f_Q^\star(x) &= \mathbb{1}\left\{ \frac{Q_{X,Y}(x,1)}{Q_{X,Y}(x,1) + Q_{X,Y}(x,2)} \geq \frac{1}{2} \right\} \\
&= \mathbb{1}\left\{ \frac{\alpha_1(x)}{\alpha_1(x) + \alpha_2(x)} \geq \frac{1}{2} \right\} \\
&= \mathbb{1}\{\alpha_1(x) - \beta_1(x) \geq 0\} = \mathbb{1}\left\{ 2a_1(x) - \frac{p' + \epsilon(1,2)(2p - 1)}{p}(a_1(x) + a_2(x)) \right\}
\end{aligned}
$$

where $a_k(x) = Q_{X,Y'}(x,k)$, $p$ and $p'$ are determined within $\mathcal{Q}(2,p)$ class. The only quantity that is not determined is $\epsilon(1,2)$. However, we notice that $f_Q^\star$ is independent of the value of $\epsilon(1,2)$ if and only if $p = (1/2, 1/2)$. For the binary classification case this concludes $\{f_Q^\star : Q \in \mathcal{Q}(K,p)\}$ is singleton if and only if $p = (1/2, 1/2)$.

## C.2 THE MULTICLASS CLASSIFICATION CASE

For $K \geq 3$ we shall prove that the Bayes decision boundary is never unique. This case is further divides in two subcases: (i) balanced $Y$ *i.e.*, $p = \mathbf{1}_K/K = \mathbf{1}/K(1, 1, \ldots, 1)^\top$ and (ii) imbalanced $Y$ *i.e.*, $p \neq \mathbf{1}_K/K$. Similar to the binary case for $\alpha_k(x) = Q_{(X,Y)}(x, k)$ and $a_k(x) = Q_{(X,Y')}(x, k)$ we have

$$\begin{bmatrix} a_1(x) \\ a_2(x) \\ \vdots \\ a_K(x) \end{bmatrix} = E \begin{bmatrix} \alpha_1(x) \\ \alpha_2(x) \\ \vdots \\ \alpha_K(x) \end{bmatrix}, \text{ where } E = \begin{bmatrix} \epsilon(1,1) & \epsilon(1,2) & \cdots & \epsilon(1,K) \\ \epsilon(2,1) & \epsilon(2,2) & \cdots & \epsilon(2,K) \\ \vdots & \vdots & \cdots & \vdots \\ \epsilon(K,1) & \epsilon(K,2) & \cdots & \epsilon(K,K) \end{bmatrix},$$

which implies

$$[\alpha_1(x), \alpha_2(x), \ldots, \alpha_K(x)]^\top = E^{-1}[a_1(x), a_2(x), \ldots, a_K(x)]^\top.$$

Note that, the vector $[a_1(x), a_2(x), \ldots, a_K(x)]^\top$ is known to us through the distribution of $P_{X,Y'}$. Additionally, we know $p'$, which is the distribution of $Y'$ and for all the distributions $Q \in \mathcal{Q}(K, p)$ the distribution of $Y$ is $p$. To establish non-identifiability we shall construct two error metrics $E_1$ and $E_2$ that are (1) stochastic (*i.e.*, has non-negative entries with column sum one), (2) has positive determinant, (3) satisfies $p' = E_1 p$ and $p' = E_2 p$ and (4) has different Bayes decision boundaries.

### C.2.1 THE BALANCED CASE

If $Y$ is class balanced ($p = \mathbf{1}_K/K$) let $E$ be the error matrix accoridng to Lemma E.2 that is invertible and satisfies $p' = Ep$. We let $E_1 = E$ and $E_2 = EP$, where the matrix $P$ is an even permutation matrix defined as $P = [e_2, e_3, e_1, e_4, \ldots, e_K]$ and $\{e_i\}_{i=1}^K$ as the standard basis of $\mathbb{R}^K$. Then, $E_1 p = Ep = p'$ and since $Pp = p$ for any permutation matrix $P$ we have $E_2 p = EPp = Ep = p'$. Defining

$$[\alpha_1(x), \alpha_2(x), \ldots, \alpha_K(x)]^\top = E_1^{-1}[a_1(x), a_2(x), \ldots, a_K(x)]^\top$$

we notice that

$$\begin{bmatrix} \tilde{\alpha}_1(x) \\ \tilde{\alpha}_2(x) \\ \tilde{\alpha}_3(x) \\ \vdots \\ \tilde{\alpha}_K(x) \end{bmatrix} = E_2^{-1} \begin{bmatrix} a_1(x) \\ a_2(x) \\ \vdots \\ a_K(x) \end{bmatrix} = P^{-1} E_1^{-1} \begin{bmatrix} a_1(x) \\ a_2(x) \\ \vdots \\ a_K(x) \end{bmatrix} = P^{-1} \begin{bmatrix} \alpha_1(x) \\ \alpha_2(x) \\ \vdots \\ \alpha_K(x) \end{bmatrix} = \begin{bmatrix} \alpha_2(x) \\ \alpha_3(x) \\ \alpha_1(x) \\ \alpha_4(x) \\ \vdots \\ \alpha_K(x) \end{bmatrix}.$$

Clearly, $\alpha(x)$ and $\tilde{\alpha}(x)$ might not yield the same decision boundary, as $\arg\max_k \alpha_k(x)$ and $\arg\max_k \tilde{\alpha}_k(x)$ may not always be the same. For example if $\alpha_2(x) = \max_{k \in [K]} \alpha_k(x)$ then with $E_1$ the optimal decision is 2 but with $E_2$ the optimal decision is 1.

### C.2.2 THE IMBALANCED CASE

For $p \neq \frac{\mathbf{1}_K}{K}$, we start with an error matrix $E_1$ as in Lemma E.2 such that $E_1 p = p'$ and $E_1$ is invertible. We let $E_2 = (1 - \delta)E_1 + \delta p' \mathbf{1}_K^T$, where $\delta > 0$ is chosen small enough such that $E_2$ remains invertible with positive determinant. Note that

$$E_2 p = (1 - \delta)E_1 p + \delta p' \mathbf{1}_K^T p = (1 - \delta)p' + \delta p' = p'.$$

Now, say $\alpha(x) = E_1^{-1} a(x)$, then

$$\tilde{\alpha}(x) = \left[(1-\delta)E_1 + \delta p' 1_K^T\right]^{-1} a(x) = \left[\frac{E_1^{-1}}{1-\delta} - \frac{\frac{\delta}{(1-\delta)^2} E_1^{-1} p' 1_K^T E_1^{-1}}{1 + \frac{\delta}{1-\delta} 1_K^T E_1^{-1} p'}\right] a(x)$$

$$= \left[I_K - \frac{\frac{\delta}{1-\delta} E_1^{-1} p' 1_K^T}{1 + \frac{\delta}{1-\delta} 1_K^T E_1^{-1} p'}\right] \frac{E_1^{-1} a(x)}{1-\delta}$$

$$= \left[I_K - \frac{\frac{\delta}{1-\delta} p 1_K^T}{1 + \frac{\delta}{1-\delta} 1_K^T p}\right] \frac{\alpha(x)}{1-\delta}$$

$$= \frac{1}{1-\delta} \left[\alpha(x) - \frac{\frac{\delta}{1-\delta} p 1_K^T \alpha(x)}{1 + \frac{\delta}{1-\delta}}\right] = \frac{1}{1-\delta} \left[\alpha(x) - \delta p\right].$$

where the second equality is obtained using the Sherman–Morrison identity on the matrix $[(1 - \delta)E_1 + \delta p' 1_K^T]^{-1}$. Since $\tilde{\alpha}(x) = \frac{1}{1-\delta}[\alpha(x) - \delta p]$ and $p \neq \frac{1_K}{K}$ the decision boundaries may not be the same as $\arg\max_k \alpha_k(x)$ and $\arg\max_k \tilde{\alpha}_k(x)$ may be different.

# D  A COMPARISON OF PEER LOSS FUNCTION AND WEIGHTED ERM

**A comparison:**  Let us consider a binary classification setup where we observe the noisy dataset $\{(x_i, y_i')\}_{i=1}^n$. In Liu & Guo (2020, Equation (5)) the peer loss function is defined as

$$\ell_{\text{peer}}(f(x_i), y_i') = \ell(f(x_i), y_i') - \ell(f(x_j), y_k'), \tag{6}$$

where $f$ is a classifier and $j \neq k$ is uniformly drawn from $[n] = \{1, \dots, n\}$. Then the peer risk for the dataset can be written as

$$\hat{L}_{\text{peer}} = \frac{1}{n} \sum_{i=1}^n \left[\ell(f(x_i), y_i') - \frac{1}{n(n-1)} \sum_{j=1}^n \sum_{k \neq j} \ell(f(x_j), y_k')\right]. \tag{7}$$

As we see in Lemma D.1, the peer loss can be simplified as

$$\hat{L}_{\text{peer}} = \frac{1}{n-1} \sum_{i=1}^n p_n(1 - y_i')\{\ell(f(x_i), y_i') - \ell(f(x_i), 1 - y_i')\}, \tag{8}$$

where, for $y \in \{0, 1\}$ the $p_n(y) = 1/n \sum_{i=1}^n \mathbb{1}\{y_i' = y\}$ is the proportion samples with noisy label $y$. This is strikingly similar to the class balanced weighted ERM with the weights $w_i = p_n(1 - y_i')$ which is defined as

$$\hat{L}_{\text{weighted}} = \frac{1}{n} \sum_{i=1}^n p_n(1 - y_i') \ell(f(x_i), y_i'). \tag{9}$$

In fact, for 0-1 loss ($\ell(f(x), y) = \mathbb{1}\{f(x) \neq y\}$) the peer risk function and the weighted empirical risks are same up-to a constant adjustment in the loss.

**A failure case of peer-loss:**  If the loss $\ell$ is not bounded, the peer loss may not be bounded below. For example, peer loss for entropy loss ($\ell(a, y) \triangleq -ay + \log\{1 + e^a\}$) simplifies to

$$\hat{L}_{\text{peer}} = -\frac{1}{n-1} \sum_{i=1}^n p_n(1 - y_i')(2y_i' - 1)f(x_i) \tag{10}$$

where $f(x)$ is the logit of prediction. To understand that the peer loss in equation 10 is may bounded below we consider logistic regression model, *i.e.*, $f(x) = x^\top \beta$, which is a simple model, and often used in many classification tasks. Then the corresponding peer loss

$$\hat{L}_{\text{peer}} = -\frac{1}{n-1} \sum_{i=1}^n p_n(1 - y_i')(2y_i' - 1)x_i^\top \beta = -\left\{\frac{1}{n-1} \sum_{i=1}^n p_n(1 - y_i')(2y_i' - 1)x_i\right\}^\top \beta \tag{11}$$

can diverge to $-\infty$ as long as $\|\beta\|_2 \to \infty$ for any $\beta$ that satisfies

$$\left\{\frac{1}{n-1} \sum_{i=1}^n p_n(1 - y_i')(2y_i' - 1)x_i\right\}^\top \frac{\beta}{\|\beta\|_2} > 0.$$

**Lemma D.1.**  *For each $i \in [n]$ let us assume that $j \neq k$ is uniformly drawn from $[n]$. Then the peer loss defined in equation 6 simplifies to*

$$\frac{1}{n-1} \sum_{i=1}^n p_n(1 - y_i')\{\ell(f(x_i), y_i') - \ell(f(x_i), 1 - y_i')\}, \tag{12}$$

*where, for $y \in \{0, 1\}$ the $p_n(y) = 1/n \sum_{i=1}^n \mathbb{1}\{y_i' = y\}$ is the proportion samples with $y_i' = y$.*

*Proof of Lemma D.1.* To simplify the peer loss we begin with he following equality.

$$
\begin{aligned}
&\frac{1}{n}\sum_{i=1}^{n}\left[\ell(f(x_i),y_i') - \frac{1}{n(n-1)}\sum_{j=1}^{n}\sum_{k\neq j}\ell(f(x_j),y_k')\right]\\
&= \frac{1}{n}\sum_{i=1}^{n}\ell(f(x_i),y_i') - \frac{1}{n(n-1)}\sum_{j=1}^{n}\sum_{k\neq j}\ell(f(x_j),y_k')\\
&= \frac{1}{n}\sum_{i=1}^{n}\ell(f(x_i),y_i') - \frac{1}{n(n-1)}\sum_{i=1}^{n}\sum_{k\neq i}\ell(f(x_i),y_k')\\
&= \frac{1}{n}\sum_{i=1}^{n}\left[\ell(f(x_i),y_i') - \frac{1}{(n-1)}\sum_{k\neq i}\ell(f(x_i),y_k')\right]
\end{aligned}
\tag{13}
$$

where the third equality is obtained simply replacing the index $j$ with the index $i$. Here, we notice that

$$
\begin{aligned}
&\ell(f(x_i),y_i') - \frac{1}{(n-1)}\sum_{k\neq i}\ell(f(x_i),y_k')\\
&= \left\{1 + \frac{1}{n-1}\right\}\ell(f(x_i),y_i') - \frac{1}{(n-1)}\sum_{k=1}^{n}\ell(f(x_i),y_k')\\
&= \frac{n}{n-1}\ell(f(x_i),y_i') - \frac{n}{n-1}\times\frac{1}{n}\sum_{k=1}^{n}\ell(f(x_i),y_k')\\
&= \frac{n}{n-1}\left\{\ell(f(x_i),y_i') - \frac{1}{n}\sum_{k=1}^{n}\ell(f(x_i),y_k')\right\}
\end{aligned}
\tag{14}
$$

and that

$$
\begin{aligned}
&\ell(f(x_i),y_i') - \frac{1}{n}\sum_{k=1}^{n}\ell(f(x_i),y_k')\\
&= \ell(f(x_i),y_i') - \frac{1}{n}\sum_{k=1}^{n}\ell(f(x_i),y_i')\mathbb{1}\{y_k' = y_i'\}\\
&\quad - \frac{1}{n}\sum_{k=1}^{n}\ell(f(x_i),1-y_i')\mathbb{1}\{y_k' = 1-y_i'\}\\
&= \ell(f(x_i),y_i') - \ell(f(x_i),y_i')p_n(y_i') - \ell(f(x_i),1-y_i')p_n(1-y_i')\\
&= p_n(1-y_i')\left\{\ell(f(x_i),y_i') - \ell(f(x_i),1-y_i')\right\}.
\end{aligned}
\tag{15}
$$

Combining equation 13, equation 14 and equation 15 we have the result. □

## E  TECHNICAL RESULTS

*Proof of lemma 2.2.* Let us define a distribution $Q$ as $q(x,y,y') = p(x,y,y')w(x,y,y')$ where $w(x,y,y') = cP(Y' = 1-y')$ for some constant $c > 0$. According to lemma E.1 instance independence assumption is still valid for $Q$, *i.e.*, $q(x,y,y') = q(x\mid y)Q(Y=y,Y'=y')$. Thus

$$
q(x\mid Y=y)Q(Y=y,Y'=y') = p(x\mid Y=y)P(Y=y,Y'=y')cP(Y'=1-y').
\tag{16}
$$

If we integrate both sides with respect to $x$ over the space $\mathcal{X}$ then we have $\int_{\mathcal{X}}q(x\mid Y=y)dx = \int_{\mathcal{X}}p(x\mid Y=y)dx = 1$ and the above equation reduces to

$$
Q(Y=y,Y'=y') = P(Y=y,Y'=y')cP(Y'=1-y').
\tag{17}
$$

Now we take a summation over $y$ in the both sides and obtain

$$
\begin{aligned}
&\sum_{y}Q(Y=y,Y'=y') = \sum_{y}P(Y=y,Y'=y')cP(Y'=1-y')\\
&\text{or, } Q(Y'=y') = P(Y'=y')cP(Y'=1-y') = cp'(1-p')
\end{aligned}
\tag{18}
$$

where $p' = P(Y'=1)$. Since, $\sum_{y'}Q(Y'=y') = 1$, from the above equation we obtain $c = \frac{1}{2p'(1-p')}$ and $w(x,y,y') = \frac{1}{2P(Y'=y')}$.

For $y\in\{0,1\}$ let us define $\alpha_y(x) = q(x\mid Y'=y')Q(Y'=y')$, $a_y = p(x\mid Y=y)P(Y=y)$, $e_0 = P(Y'=1\mid Y=0)$, $e_1 = P(Y'=0\mid Y=1)$ and notice that

$$
\begin{aligned}
\alpha_1(x) &= q(x\mid Y'=1)Q(Y'=1) = q(x,1,1) + q(x,0,1)\\
&= \frac{p(x,1,1)}{2p'} + \frac{p(x,0,1)}{2p'}\\
&= \frac{p(x\mid Y=1)P(Y=1)P(Y'=1\mid Y=1)}{2p'} + \frac{p(x\mid Y=0)P(Y=0)P(Y'=1\mid Y=0)}{2p'}\\
&= \frac{a_1(x)1/2(1-e_1)}{2p'} + \frac{a_0(x)1/2e_0}{2(1-p')} = \frac{a_1(x)(1-e_1)}{4p'} + \frac{a_0(x)e_0}{4p'}.
\end{aligned}
\tag{19}
$$

Similarly, we obtain

$$
\alpha_0(x) = \frac{a_1(x)e_1}{4(1-p')} + \frac{a_0(x)(1-e_0)}{4(1-p')}.
\tag{20}
$$

Taking the differences between equation 19 and equation 20 we obtain

$$\alpha_1(x) - \alpha_0(x) = \frac{a_1(x)}{2}\left(\frac{1-e_1}{2p'} - \frac{e_1}{2(1-p')}\right) + \frac{a_0(x)}{2}\left(\frac{e_0}{2p'} - \frac{1-e_0}{2(1-p')}\right). \tag{21}$$

Here, we use

$$p' = P(Y'=1) = \frac{1}{2}(1-e_1) + \frac{1}{2}e_0 \implies 2p' = 1-e_1+e_0, \text{ and } 2(1-p') = 1-e_0+e_1 \tag{22}$$

in the above equation and obtain

$$\begin{aligned}
\alpha_1(x) - \alpha_0(x) &= \frac{a_1(x)}{2}\left(\frac{1-e_1}{2p'} - \frac{e_1}{2(1-p')}\right) + \frac{a_0(x)}{2}\left(\frac{e_0}{2p'} - \frac{1-e_0}{2(1-p')}\right) \\
&= \frac{a_1(x)}{2}\left(\frac{1-e_1}{1-e_1+e_0} - \frac{e_1}{1-e_0+e_1}\right) + \frac{a_0(x)}{2}\left(\frac{e_0}{1-e_1+e_0} - \frac{1-e_0}{1-e_0+e_1}\right) \\
&= \frac{a_1(x)}{2}\left(\frac{1-e_1}{1-e_1+e_0} - \frac{e_1}{1-e_0+e_1}\right) + \frac{a_0(x)}{2}\left(1 - \frac{1-e_1}{1-e_1+e_0} - 1 + \frac{e_1}{1-e_0+e_1}\right) \\
&= \frac{a_1(x)}{2}\left(\frac{1-e_1}{1-e_1+e_0} - \frac{e_1}{1-e_0+e_1}\right) + \frac{a_0(x)}{2}\left(-\frac{1-e_1}{1-e_1+e_0} + \frac{e_1}{1-e_0+e_1}\right) \\
&= \frac{a_1(x)-a_0(x)}{2}\left(\frac{1-e_1}{1-e_1+e_0} - \frac{e_1}{1-e_0+e_1}\right) = \frac{a_1(x)-a_0(x)}{2}\left(\frac{1-e_1-e_0}{(1-e_1+e_0)(1-e_0+e_1)}\right).
\end{aligned} \tag{23}$$

Since, $1-e_1+e_0, 1-e_0+e_1 \geq 1-e_1-e_0 > 0$ we have $\frac{1-e_1-e_0}{(1-e_1+e_0)(1-e_0+e_1)} > 0$. Now we see that $a_1(x) \geq a_0(x)$ if and only if $\alpha_1(x) \geq \alpha_0(x)$. Now, noticing that (1) $P(Y=1 \mid X=x) \geq \frac{1}{2}$ if and only if $a_1(x) \geq a_0(x)$, and (2) $Q(Y'=1 \mid X=x) \geq \frac{1}{2}$ if and only if $\alpha_1(x) \geq \alpha_0(x)$ we have

$$\{x : Q(Y'=1 \mid X=x) \geq \frac{1}{2}\} = \{x : P(Y=1 \mid X=x) \geq \frac{1}{2}\}.$$

This implies the Bayes decision boundaries for $Y'$ on the $q(x,y,y')$ distribution and for $Y$ on $P$ distribution are same and

$$f_P^\star(x) = \arg\min_{f \in \mathcal{F}} \mathbb{E}_P[p'(1-Y')\ell(f(X),Y')]. \tag{24}$$

$\square$

**Lemma E.1** (Reweighting of the noisy labels). *Say $P \in \mathcal{Q}(K,p)$ satisfying the conditional independence property that $P_{Y'|Y,X} = P_{Y'|Y}$ and $p' = P_y$. Then the sample can be reweighted to obtain class balanced $Y'$ that is $p' = P_y = \frac{1_k}{k}$ while satisfying the conditional independence $X \perp Y' \mid Y$ condition for the reweighted distribution.*

*Proof.* After the reweighting

$$q(x,y,y') = p(x,y,y')\frac{1}{kp(y')} = p(x|y)\frac{p(y,y')}{kp(y')} = \frac{p(x|y)p(y|y')}{k}$$

the $Y'$ gets class balanced, as seen below.

$$\begin{aligned}
q(y') &= \sum_y \int_{x,y} q(x,y,y')dx = \frac{1}{k}\sum_y \int_{x,y} p(x|y)p(y|y')dx \\
&= \frac{1}{k}\sum_y \left(\int_x p(x|y)dx\right)p(y|y') = \frac{1}{k}\sum_y p(y|y') = \frac{1}{k}
\end{aligned}$$

For such reweighting the conditional independence of $X \perp Y' \mid Y$ still is satisfied for $q$:

$$q(x|y,y') = \frac{q(x,y,y')}{q(y,y')} = \frac{\frac{1}{k}p(x|y)p(y|y')}{\frac{1}{k}p(y|y')} = p(x|y).$$

$\square$

**Lemma E.2.** *Let $p$ and $p'$ be any probability vectors on the space $[K]$. Let us assume that the entries of $p$ and $p'$ are all positive then there exists a matrix $E$ such that (1) its entries are non-negative, (2) the column sums are all one, (3) the determinant is positive, and (4) $p' = Ep$.*

*Proof of lemma E.2.* Let us assume that $P \in \mathbb{R}^{K \times K}$ is the permutation matrix that reorders the $p' - p$ in a decreasing fashion, *i.e.*, the entries of $P(p'-p)$ are decreasing. Note that $Pp$ and $Pp'$ are

still probability vectors that have positive entries. Let us define $K_1 = \max\{k : [Pp]_k \leq [Pp']_k\}$. Now we define our stochastic matrix $\tilde{E} = [\tilde{e}_{ij}]_{i,j\in[K]}$ as the following.

$$\tilde{e}_{i,j} = \begin{cases} 1 & \text{if } i = j \leq K_1, \\ [Pp']_i/[Pp]_i & \text{if } i = j \geq K_1 + 1, \\ \dfrac{[P(p'-p)]_i[P(p-p')_j]}{[Pp]_j \sum_{k=1}^{K_1}[P(p'-p)]_k} & \text{if } i \leq K_1,\ j \geq K_1+1, \\ 0 & \text{elsewhere} \end{cases} \tag{25}$$

Note that, for $i \leq K_1$

$$\sum_{j=1}^K \tilde{e}_{ij}[Pp]_j = \tilde{e}_{ii}[Pp]_i + \sum_{j\geq K_1+1} \tilde{e}_{ij}[Pp]_j$$
$$= 1 \times [Pp]_i + \sum_{j\geq K_1+1} \frac{[P(p'-p)]_i[P(p-p')_j]}{[Pp]_j \sum_{k=1}^{K_1}[P(p'-p)]_k} \times [Pp]_j$$
$$= [Pp]_i + [P(p'-p)]_i \times \frac{\sum_{j=K_1+1}^K [P(p-p')_j]}{\sum_{k=1}^{K_1}[P(p'-p)]_k}$$

If it holds (to be established later)

$$\sum_{j=K_1+1}^K [P(p-p')_j] = \sum_{k=1}^{K_1}[P(p'-p)]_k \tag{26}$$

then for $i \leq K_1$ we have

$$\sum_{j=1}^K \tilde{e}_{ij}[Pp]_j = [Pp']_i .$$

Additionally for $i \geq K_1 + 1$ we have

$$\sum_{j=1}^K \tilde{e}_{ij}[Pp]_j = e_{ii}[Pp]_i = \frac{[Pp']_i}{[Pp]_i} \times [Pp]_i = [Pp']_i .$$

This implies $Pp' = \tilde{E}Pp$ or $p' = P^{-1}\tilde{E}Pp$. Since the permutation matrix $P$ is also orthogonal, we have $P^{-1}\tilde{E}P = P^\top\tilde{E}P$. We define $E = P^\top\tilde{E}P$ Then $p' = Ep$, which verifies (4). Note that $E$ is obtained simply by permuting the rows and columns of $\tilde{E}$ according to the permutation matrix $P^\top$.

Clearly, (1) is satisfied because the entries of $\tilde{E}$ and hence of $E$ are non-negative.

We verify (2) for $\tilde{E}$, which implies the same for $E$, because

$$\mathbf{1}^\top E = \mathbf{1}^\top P^\top \tilde{E} P = \mathbf{1}^\top \tilde{E} P = \mathbf{1}^\top P = \mathbf{1}^\top .$$

We notice that for $j \leq K_1$ the $j$-th column is simply $e_j$ (the $j$-th canonical basis of $\mathbb{R}^K$) which has column sum one. If $j \geq K_1 + 1$ then the column sum is

$$\sum_{i=1}^K \tilde{e}_{i,j} = \sum_{i\leq K_1} \frac{[P(p'-p)]_i[P(p-p')_j]}{[Pp]_j \sum_{k=1}^{K_1}[P(p'-p)]_k} + \frac{[Pp']_j}{[Pp]_j} = \frac{\left(\sum_{i\leq K_1}[P(p'-p)]_i\right)[P(p-p')_j]}{[Pp]_j \sum_{k=1}^{K_1}[P(p'-p)]_k} + \frac{[Pp']_j}{[Pp]_j}$$
$$= \frac{[P(p-p')_j]+[Pp']_j}{[Pp]_j} = 1 .$$

To verify (3) we first notice that $\tilde{E}$ and $E = P^\top\tilde{E}P$ have the same determinant, since $P$ is an orthogonal matrix. So, we shall only prove that $\tilde{E}$ have positive determinant. Now, we notice that $\tilde{E}$ is a upper triangular matrix, whose determinant is equal the product of the diagonal entries, i.e., $\prod_{i\geq K_1+1} [Pp']_i/[Pp]_i$. Since $Pp$ and $Pp'$ have positive entries, the determinant is positive.

It remains to verify equation 26. Since $\sum_{i=1}^K [P(p'-p)]_i = 0$ we have

$$\sum_{i=1}^{K_1}[P(p'-p)]_i = -\sum_{i=K_1}^K [P(p'-p)]_i = \sum_{i=K_1}^K [P(p-p')]_i$$

which verifies equation 26. $\qquad\square$

