# OpenReview forum: "Bayes classifier cannot be learned from noisy responses with unknown noise rates"
_ICLR.cc/2023/TinyPapers — Submitted to Tiny Papers @ ICLR 2023_

### Official Review · Reviewer_syRC · 2023-03-28

**Confidence:** 3

**Summary Of Contributions:**

This paper focuses on classification with noisy labels and the difficulty of dealing with the unknown distribution of label noise. The authors demonstrate that in most cases, the Bayes decision rule is unidentified when dealing with noisy labels, indicating that knowledge of the noise distribution is necessary. However, they also propose a simple method for learning the Bayes decision rule without prior knowledge of the noise distribution, as long as it can be identified.

**Rating:**

Clear, Correct, and Reproducible (CCR): a submission which meets the reviewing criteria

**Strengths And Weaknesses:**

Strengths:

- The paper is well-written and presents a thorough investigation of the identifiability of the Bayes classifier in classification scenarios with noisy labels.
- The authors provide a clear explanation of the problem and their findings.

Weaknesses:

- The paper's strengths include its theoretical rigor and the development of a simple algorithm to learn the Bayes decision rule in certain cases. However, the paper's narrow focus on the Bayes classifier may limit its practical applicability, and it would benefit from more discussion of other approaches and methods for classification with noisy labels.

**Suggested Changes:**

Suggested changes:

- The authors could expand their discussion to include other approaches and methods for classification with noisy labels beyond the Bayes classifier, as well as potential applications and real-world examples of the problem.
- Additionally, showing experimental results on a simple dataset would improve the paper. I would recommend two experiments: 1) demonstrating that the proposed method works well regardless of the label noise distribution in the balanced binary classification problem, and 2) showing the failure of the Bayes classifier in the imbalanced binary classification problem.

---

> ### Author Response · Authors · 2023-04-13
> **Response to Reviewer syRC**
>
> Thank you for reviewing our paper and insightful comments. Please see our responses below.
>
> **Other approaches for classification with noisy labels beyond the Bayes classifier.** The correct identification of the Bayes classifier is crucial for any real-world classification problem. The Bayes classifier, which yields the highest classification accuracy on a test dataset, is a fundamental quantity of interest for classification problems, and various classification methods, such as decision trees, random forests, boosting, and SVM, consistently estimates it [1]. However, if the Bayes classifier is inaccurately identified, none of these classification methods can generate a statistically consistent classifier for the true labels [2].
>
>
> **Experimental results on a simple dataset.** Thank you for the suggestion. Please see our synthetic experiments in Appendix B, that demonstrates *the consistency of our class balancing reweighted approach for balanced binary classification* problem and *its inconsistency in an imbalanced case.*
>
>
> ---
> ## References
>
> [1] Bartlett, P. L., Jordan, M. I., & McAuliffe, J. D. (2006). Convexity, classification, and risk bounds. Journal of the American Statistical Association, 101(473), 138-156
>
> [2] Lugosi, G., & Vayatis, N. (2004). On the Bayes-risk consistency of regularized boosting methods. The Annals of statistics, 32(1), 30-55.

---

### Official Review · Reviewer_42Pu · 2023-03-30

**Confidence:** 3

**Summary Of Contributions:**

The authors show that in multiclassification and unbalanced binary settings with noisy labels, Bayes decision rules are unidentifiable.

**Rating:**

High Potential (HP): a submission which meets the reviewing criteria and has potential to make an impact on the field

**Strengths And Weaknesses:**

The paper is well-written and assesses a meaningful problem. The authors motivate their contributions very clearly and concisely.  The proofs are well-supported and written which improves understandability and engagement with the concepts.

Given that it's a tiny paper, I think the authors did an excellent job of putting the key ideas forward.
Although most content is in the appendix, the authors provided key ideas and theorem statements in the main body which made it easier to follow. I believe the full paper will reduce the work posted in the appendix, but in general, I find the presentation excellent.

The authors do answer fundamental questions, however, they don't provide a nice link to related work. The authors should improve their assessment of related work and highlight the gaps they novelly bridge in the literature.

**Suggested Changes:**

The authors should improve their assessment of related work and highlight the gaps they novelly bridge in the literature.

Not suggestions, questions:

- I am curious about cases where y is continuous and when data is a mix of categorical and non-categorical variable.

- Would identifiability be achieved if one maps the balanced multi-classification model to a balanced paired binary classification model?

---

### Meta-Review · Area_Chair_8ktT · 2023-04-07

**Recommendation:** Invite to present
**Confidence:** 5

**Metareview:**

This is an interesting work that shows
- One can not easily relax the assumption that label noise distribution is known for training a noise-robust classifier.
- However, this assumption can be relaxed in a balanced binary classification setting but not in general settings where classification label space is huge.

I am not sure how the researcher will use this finding but is worth noting.

**Summary:**

The main message of the paper is that the availability of label noise distribution is not possible to relax to train a classifier in label noise settings. This is theoretically shown that the only cases of binary classification on balanced datasets where this requirement can be relaxed.

**Reason For Not Giving A Higher Recommendation:**

Though the paper is well-written and theoretically proves guarantees, it lacks empirical evidence. Authors are suggested to show the usefulness of their findings on a simple dataset to start with.

**Reason For Not Giving A Lower Recommendation:**

NA

---

### Decision · Program_Chairs · 2023-04-10

Invite to present